

# Block-based compressive sensing in deep learning using AlexNet for vegetable classification

Indrarini Dyah Irawati[1], Gelar Budiman[2], Sofia Saidah[2], Suci Rahmadiani[2] and Rohaya Latip[3]

[1] School of Applied Science, Telkom University, Bandung, West Java, Indonesia
[2] School of Electrical Engineering, Telkom University, Bandung, West Java, Indonesia
[3] Department of Communication Technology and Network/Faculty of Computer Science and Information Technology, Universiti Putra Malaysia, Serdang, Selangor, Malaysia

## ABSTRACT

Vegetables can be distinguished according to differences in color, shape, and texture. The deep learning convolutional neural network (CNN) method is a technique that can be used to classify types of vegetables for various applications in agriculture. This study proposes a vegetable classification technique that uses the CNN AlexNet model and applies compressive sensing (CS) to reduce computing time and save storage space. In CS, discrete cosine transform (DCT) is applied for the sparsing process, Gaussian distribution for sampling, and orthogonal matching pursuit (OMP) for reconstruction. Simulation results on 600 images for four types of vegetables showed a maximum test accuracy of 98% for the AlexNet method, while the combined block-based CS using the AlexNet method produced a maximum accuracy of 96.66% with a compression ratio of 2×. Our results indicated that AlexNet CNN architecture and block-based CS in AlexNet can classify vegetable images better than previous methods.

Corresponding author
Indrarini Dyah Irawati,
indrarini@telkomuniversity.ac.id

## INTRODUCTION

Indonesia is one of the world's top producers of horticultural crop commodities, especially vegetables. Vegetables produced by Indonesian farmers are mostly used to meet daily household needs, but some are industrially processed for various processed products that are even exported abroad. Vegetables are foodstuffs that contain several vitamins and minerals and provide many benefits for the health of the human body. In addition, growing vegetables gives farmers a high chance of increasing their profits in Indonesia. Therefore, there is a need for a digital approach to classify vegetables more easily.

Technology is very influential in everyday life because individuals can carry out activities more effectively and efficiently (*Zeng, 2017*). Previous research (*Chauhan, Ghanshala & Joshi, 2018*) used CNN to detect and recognize objects in images by imitating the image recognition system in the human visual cortex. Another study (*Hameed, Chai & Rassau, 2020*) utilized CNN for object categorization with promising results. CNN

architecture was also studied for sound/music classification in research by *Doukhan & Carrive (2017)*. GoogLeNet architecture has also been used to monitor freshness in bananas, and the results showed that the model could detect banana freshness with an accuracy of 98.92%, which was higher than the human detection rate. The fruits tested in this study were limited to bananas and strawberries (*Ni et al., 2020*). Sorting tomatoes conventionally can be costly, unproductive, and the reliability is uncertain. Therefore, this study aimed to design a powerful AI-based controller for the process of sorting tomatoes into three categories (namely raw, ripe, and defective (ripe and rotten)), as well as provide designs for a cost-effective tomato sorting machine. An information processing approach with reliable estimation capabilities that works based on the workings of the biological nervous system in human brain cells is known as an artificial neural network (ANN). In agriculture, ANN can be used to detect oil palm plants infected with Ganoderma through remote sensing based on the Unmanned Image Aerial Vehicle (UAV) (*Ahmadi et al., 2022*). Research by *Haggag et al. (2019)* developed a control algorithm based on CNN-ANN with 100% classification performance for all classes. Classification results reach 100% for raw and ripe data and 90% for mature and defective (ripe and rotten). *Pratondo, Elfahmi & Novianty (2022)* developed a model to classify the Zingiberaceae species from images that were directly photographed with a cellphone camera. The pre-trained visual geometri group (VGG-19) and Inception V3 with ImageNet weights using transfer learning simulations resulted in accuracies of 92.43% and 94.29%, respectively. Identifying weeds in plantation vegetable crops is quite challenging because there are many species of weeds; therefore, deep learning technology and image processing technology are used to identify them. CenterNet architecture is trained to detect vegetables with precision, recall, and F1-score values of 95.6%, 95%, and 95.3% respectively (*Jin, Che & Chen, 2021*).

The classification of vegetables and circular fruit, namely apples, lemons, oranges, pomegranates, tomatoes, and colored peppers, is carried out using the GoogLeNet architecture. The results of this study obtained a GoogLeNet training accuracy of 96.88% and test accuracy of 96% (*Yuesheng et al., 2021*). Classification of images of fruits and vegetables in supermarkets is a complex problem. CNN have shown promising results for object classification. Transfer learning and ensemble techniques can be used to overcome the scarcity of datasets. Accuracy results were obtained using the GoogLeNet architecture in 95% training, 93% testing, 95% ensembles, and 96% training accuracy results, 94% testing, and 98% ensembles obtained by MobileNet (*Hameed, Chai & Rassau, 2020*). Detection of organic and non-organic fruits and vegetables using machine learning methods was developed using a combination of spectroscopy. The best accuracy was 95.1%, which was obtained from the artificial neural network (ANN) architectural model (*Natarajan & Ponnusamy, 2021*).

Research by *Li, Fu & Yu (2017)* conducted a comparison of the Deep-CNN (DCNN), ANN, and TMPL methods with high-resolution remote sensing imagery to identify oil palm trees. The results of this study showed that CNN obtained the highest accuracy value, 92–97% in four regions, then ANN with 83–85%, and then TMPL with 70–77%. An intra-class vegetable recognition system used the CNN method with 24 categories of vegetables. The accuracy of the vegetable recognition rate obtained in this study was 95.50%

(*Sudharshan Duth & Jayasimha, 2020*). Another study developed by *Litvak, Divekar & Rabaev (2022)* provided the Urban Planter dataset for classifying plant species consisting of 1,500 images with 15 types of houseplants. This study also tested DCNN different configurations and pre-trained models and web applications to evaluate the architectural model. Deep learning-based real-time vegetable recognition system has been used for agricultural robots. Comparative experiments showed that the recognition system could effectively identify seven different types of vegetables (the average AP can reach 87.89%, and the detection speed is up to 38FPS) (*Zheng et al., 2019*). DCNN-based vegetable image classification was tested by *Ahmed, Mahmud Mamun & Zaman Asif (2021)*. Comparison tests of several CNN architectures such as VGG16, MobileNet, InceptionV3, and ResNet were carried out using transfer learning scenarios to get the optimum architecture built upon the accuracy and effectiveness of the new image data set. Fruit and vegetable classification using an android application with a deep learning method is easy to access. In *Agrawal et al. (2021)*, the research was split into two parts: the first part explored the model through its training and development, and the second part was developed to obtain a simple model through trimming techniques so that the process was lighter but still maintained accuracy. Previous research also used AlexNet architecture in classifying the types of vegetables, which consisted of five classes of vegetables. The result of the accuracy test in this study was 92.1% (*Zhu et al., 2018*). The same study was also conducted using three classes of vegetables and the You Only Look Once algorithm and succeeded in predicting with an accuracy of 61.6% (*Sachin et al., 2019*).

The need for storage space and image data transmission is often an obstacle in the classification process because storage requires sufficient space. In contrast, the image data transmission process requires considerable time and bandwidth. The application of CS on image data can minimize the delivery time of image data and requires less storage space. The research combining CS procedure and deep learning is rare. The novelty of our method is the CS contribution to the system with the deep learning procedure. In this study, the development of research was carried out by applying CS to the CNN, thus the main contributions of this article consist of (1) efficient storage usage, (2) delivery time reduction, (3) The accuracy of testing is tremendous although the data quality is reduced and (4) data bandwidth is reduced before training and testing process obtaining an efficient data transmission.

## MATERIALS AND METHODS

The dataset used in this study is a secondary dataset obtained from Kaggle in JPG format as shown in Fig. 1 (*Ahmed, Mahmud Mamun & Zaman Asif, 2021*). The data consisted of four classes (image data of broccoli, carrot, potato, and radish), and each class was made up of RBG images. The dataset varied and was taken from different perspectives so that the CNN algorithm could recognize objects and learn more about the object under study.

Figure 2 shows the block diagram of the proposed method. The input was an RGB vegetable image during the preprocess stage. The original image was resized to $256 \times 256$ pixels. The next step in the process was block splitting, which made the image into blocks with a size of $8 \times 8$.

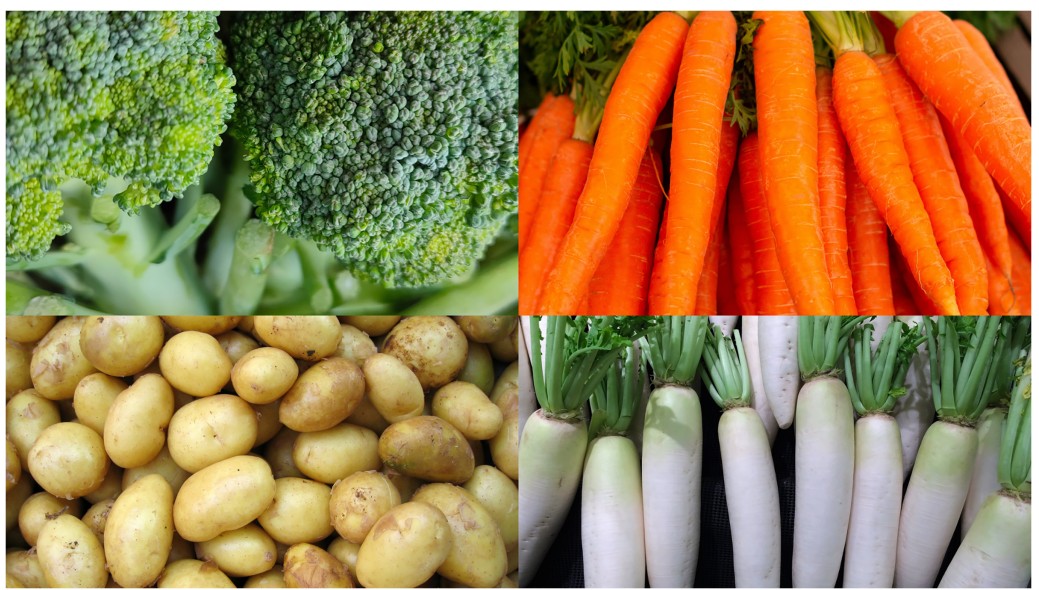

**Figure 1 Vegetable dataset examples. (A) Broccoli. (B) Carrots. (C) Potatoes. (D) Radishes (Kaggle, *Ahmed, Mahmud Mamun & Zaman Asif, 2021*).**

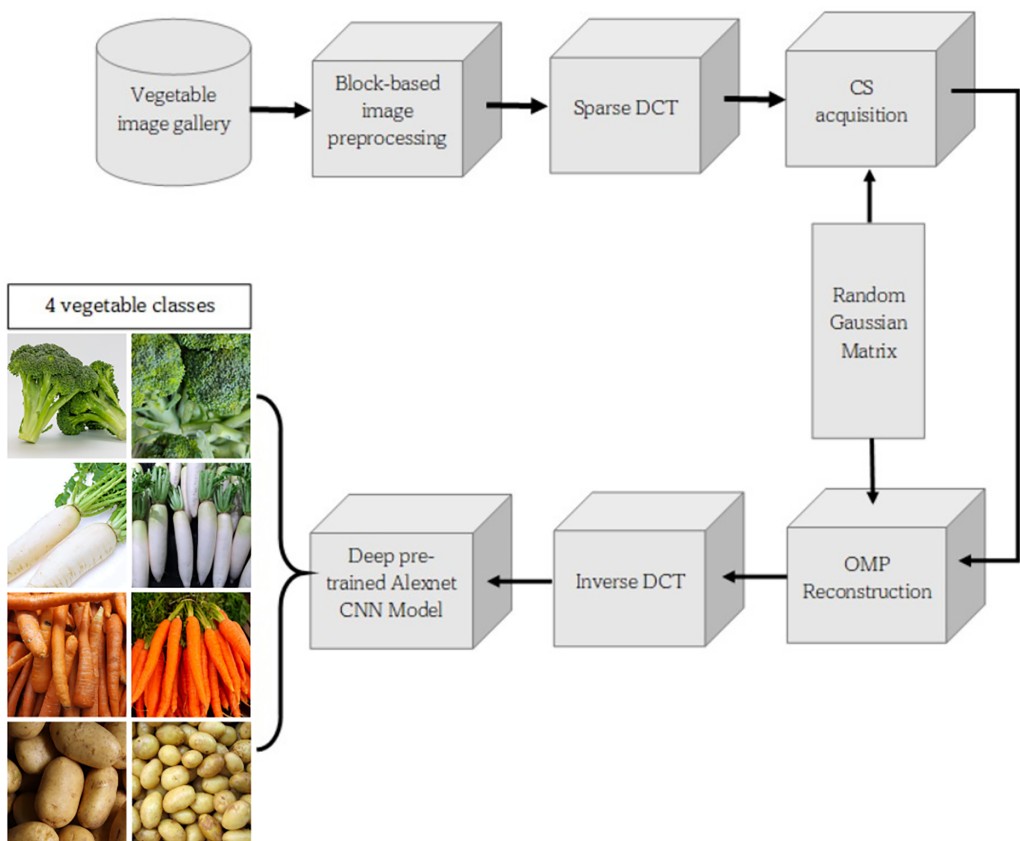

**Figure 2 Block diagram of proposed method (Kaggle, *Ahmed, Mahmud Mamun & Zaman Asif, 2021*).**

In the CS stage, the sparse features of signals in a transform domain were used to perform non-uniform sampling of signals at a sample rate significantly lower than Nyquist's sampling frequency, allowing the sampling and compression processes to be merged (*Ouyang, Hu & Hou, 2019*). The sparsing process was carried out on the $8 \times 8$ block. The next step was to convert the two-dimensional or $8 \times 8$ signal into a $64 \times 1$ column vector.

CS consists of two stages: compression and reconstruction. For compression, the signal is acquired with a Gaussian measurement matrix (A) measuring $M \times 64$ so that a matrix of $M \times 1$ size is obtained. While the reconstruction process aims to restore the original signal, OMP is one of the greedy and iterative reconstruction algorithms that performs calculations repeatedly to get optimal results (*Usman, 2017*). The next stage is transforming the signal from the frequency domain to the spatial domain using IDCT to get the original signal (*Irawati et al., 2022*).

CNN is a neural network commonly used in image data, and consists of neurons with weight, bias, and activation functions (*Doukhan & Carrive, 2017*). It is an extension of the multilayer perceptron (MLP). The CNN architecture consists of two main parts, feature learning and classification. In the first stage of feature learning, the convolution stage uses a kernel of a certain size. The number of kernels used depends on the number of features generated from the calculation. After the activation function using a rectifier linear unit (ReLU) and the activation function process is exited, the pooling process is carried out. This process is repeated several times until it is obtained. This feature map is sufficient to continue into the classification section.

In the LSVRC-2010 ImageNet contest, AlexNet was proposed to conduct classification on 1.2 million high-resolution photos. Non-saturating neurons and a GPU implementation of convolution operations made training go faster (*Sustika et al., 2018*). The suggested neural network system came in first place among participants at the ILSVRC2012. A test error rate of 13.3% was achieved by the networks. AlexNet has been utilized for a variety of tasks since its initial publication, including object identification (*Drayer & Brox, 2014*), picture segmentation (*Long, Shelhamer & Darrell, 2015*), and video categorization (*Ng et al., 2015*).

The block-based CS algorithm in this study is described in two steps: the training and testing process. The training algorithm is described in detail according to the following steps:

1. Read an RGB image and resize to a certain resolution $(N \times N)$ obtaining $x(n, m)$.
2. Generate random normal distributed $A \, (M \times L)$ matrix for CS acquisition where $L = B^2$, $B$ is the size of segmented image and $M \ll L$.
3. Apply segmentation of the image into $B \times B$ pixels obtaining $x_i \, (B \times B)$ where $i$ is a block number of the block-based image.
4. Transform $x_i$ into DCT domain by 2-D DCT obtaining $X_i \, (B \times B)$.
5. Reshape 2-D matrix $X_i$ into 1-D vector obtaining $v_i$ with size $B^2 \times 1$.
6. Apply CS acquisition $Av_i$ obtaining $Y_i$ with size $M \times 1$.

7. Reconstruct $Y_i$ by OMP using $A$ obtaining $\widehat{\mathbf{v}}_i$ with size $B^2 \times 1$.

8. Reshape 1-D vector $\widehat{\mathbf{v}}_i$ into 2-D matrix $\widehat{\mathbf{X}}_i$ with size $B \times B$.

9. Apply 2-D IDCT to $\widehat{\mathbf{X}}_i$ obtaining $\widehat{\mathbf{x}}_i$ with size $B \times B$.

10. Repeat steps 4–9 until all blocks are processed.

11. Combine all blocks from step 10 obtaining the reconstructed image $\hat{x}(n, m)$ with size $N \times N$.

12. Repeat steps 1–11 until we have 400 reconstructed images for training. These images are ready to go to the deep learning training stage.

13. Train the dataset consisting of 400 images using the AlexNet technique in CNN model to obtain the trained network. Save the trained network.

While the steps for the testing phase are as follows:

1. Read an RGB image and resize to the certain resolution $(N \times N)$ obtaining $x(n, m)$.

2. Generate random normal distributed $A(M \times L)$ matrix for CS acquisition where $L = B^2$, $B$ is the size of segmented image and $M \ll L$.

3. Apply segmentation of the image into $B \times B$ pixels obtaining $x_i$ $(B \times B)$ where $i$ is a block number of the block-based image.

4. Transform $x_i$ into DCT domain using 2-D DCT to obtain $X_i$ $(B \times B)$.

5. Reshape 2-D matrix $X_i$ into 1-D vector obtaining $v_i$ with size $B^2 \times 1$.

6. Apply CS acquisition $Av_i$ obtaining $Y_i$ with size $M \times 1$.

7. Reconstruct $Y_i$ by OMP using $A$ obtaining $\widehat{\mathbf{v}}_i$ with size $B^2 \times 1$.

8. Reshape 1-D vector $\widehat{\mathbf{v}}_i$ into 2-D matrix $\widehat{\mathbf{X}}_i$ with size $B \times B$.

9. Apply 2-D IDCT to $\widehat{\mathbf{X}}_i$ obtaining $\widehat{\mathbf{x}}_i$ with size $B \times B$.

10. Repeat steps 4–9 until all blocks are processed.

11. Combine all blocks from step 10 to obtain the reconstructed image $\hat{x}(n, m)$ with size $N \times N$.

12. Load the trained network and apply testing to the reconstructed image using AlexNet technique and the trained network. Save the testing result.

13. Repeat steps 1–12 for the next testing of the reconstructed image. Calculate the deep learning performance of all testing images according to accuracy, precision, recall and F1-score.

The training and testing process is also illustrated in Fig. 3.

The confusion matrix is one of the parameters used to evaluate the classification method (*Sokolova & Lapalme, 2009*). At the testing stage, the stored training data will be processed. The test will be presented in the form of a confusion matrix to obtain precision, recall, F1-scores, and accuracy as shown in Eqs. (1)–(4). The true negative (TN) value is the amount of negative data that is correctly classified by the system. True positive (TP) is the amount of positive data that is correctly classified by the system. False positive (FP) is the

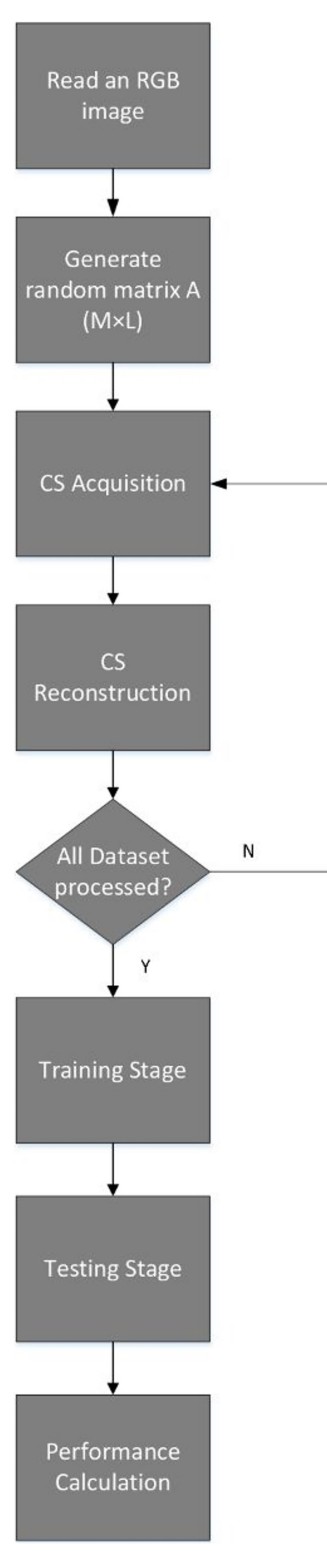

Figure 3 Training and testing flowchart.               

number of positive data that is incorrectly classified by the system. False negatives (FN) are the number of negative data that the system incorrectly classifies.

$$Precision = \frac{TP}{TP + FP} \tag{1}$$

$$Recall = \frac{TP}{TP + FN} \tag{2}$$

$$F1-score = \frac{2 * Precision * Recall}{Precision + Recall} \tag{3}$$

$$Accuracy = \frac{TP + TN}{TP + TN + FP + FN} \tag{4}$$

The compression performance measurement is expressed in a compression ratio (CR), which is the ratio between the original signal and the compressed signal, as shown in Eq. (5) (*Irawati, Suksmono & Edward, 2018*).

$$CR = \frac{uncompressed\ size}{compressed\ size} \tag{5}$$

## RESULTS

In this study, the system model designed for dataset testing uses the AlexNet architecture with predetermined input parameters as shown in Table 1. The dataset was divided into three out of 600 images: 70% training data with 420 images, 10% validation data with 60 images, and 20% test data with 120 randomly selected images. The dataset used for the CNN process had been reconstructed during the CS process. The programming platform used in this research was Google Collaboratory, starting from the pre-processing, compressive sensing, and classification stages.

The accuracy and loss performance of the training and validation process using the AlexNet method are shown in Fig. 4, while the combination method of block-based CS on AlexNet with a CR of 2× is shown in Fig. 5. Figure 4 shows that the result of the training accuracy was 98% while the validation accuracy was 96%. The result of the training loss was 14% and validation loss was 15%. In Fig. 5, the results showed that the training accuracy was 97% while the validation accuracy was 86%. Loss training performance was 11% and validation loss was 36%.

In the testing phase, 20% of the images from the dataset were used. Figure 6 shows that 112 images were correctly predicted according to their class. This condition shows that the application of block-based CS on the CNN model using the AlexNet architecture is promising for classifying vegetables such as carrots, broccoli, radishes, and potatoes. The description of the confusion matrix on the model's ability to predict different types of vegetables is shown in Table 2.

In this article, we tested the effect of several CRs on the system performance when using AlexNet only *vs* the combination of blocked-based CS and AlexNet. There were five CRs: 32, 16, 8, 4, and 2. Table 3 shows the performance parameters based on the F1-score, recall, precision, testing accuracy, and testing loss. Based on these results, we determined that the

**Table 1 Predetermined input parameter.**

| Parameters | Value |
|---|---|
| Resize | 256 × 256 |
| CS ratio | 6.25% |
| Optimizer | SGD |
| Epoch | 100 |
| Batch size | 64 |
| Dropout | 0.4 |
| Learning rate | 0.001 |

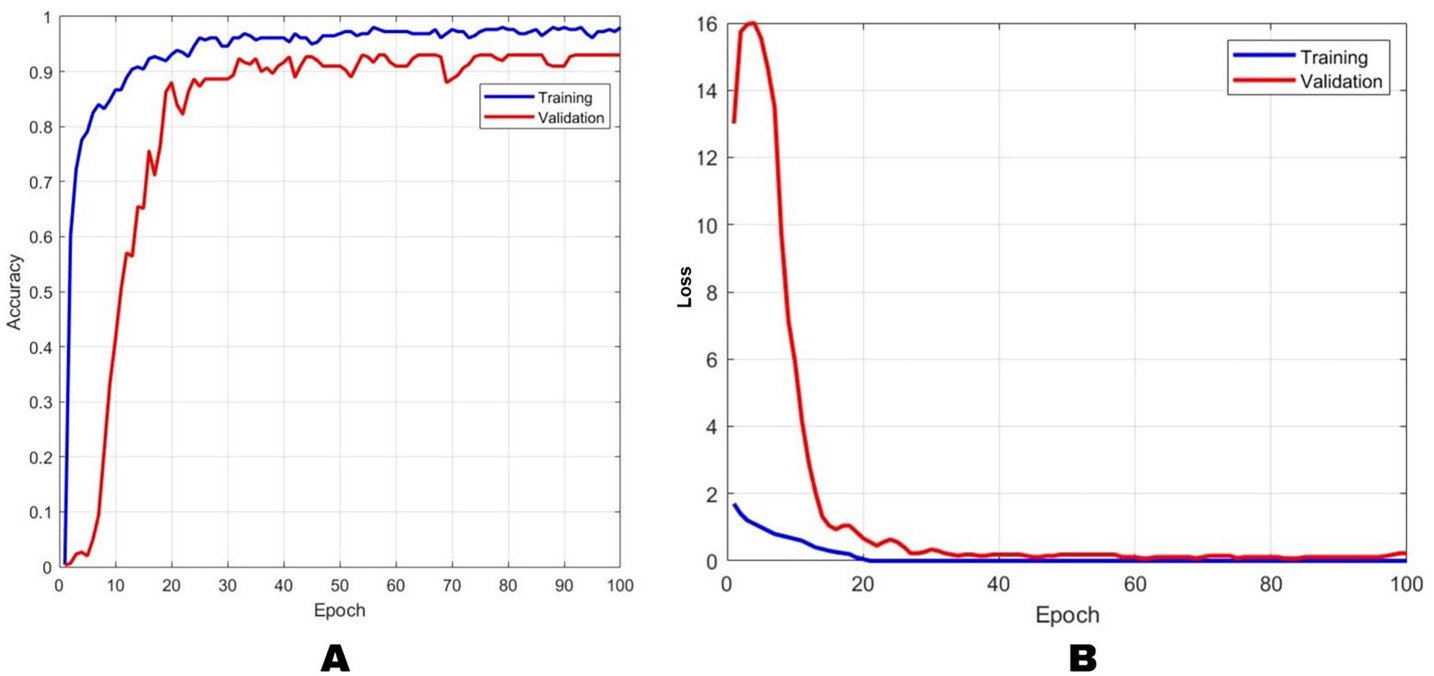

**Figure 4 The results of training and validation on the AlexNet method for (A) accuracy and (B) loss.** The blue line indicates the training results, and the orange line represents the testing results.

smaller the CR, the higher the precision, recall, F1, and test accuracy values, and the value of testing loss was directly proportional to CR. There is a trade-off between accuracy and CR. When the CR is getting better or higher, the accuracy will be lower. This is in accordance with the quality of the image which gets worse when the CR is higher, so that the deep learning system will find it more difficult to get training accuracy. However, the lower the accuracy, the higher the CR, which has a good impact on the efficiency of the process. In this case we choose a moderate value between accuracy and CR so that both provide benefits or a win-win solution depending on the performance focus, whether to choose CR or accuracy.

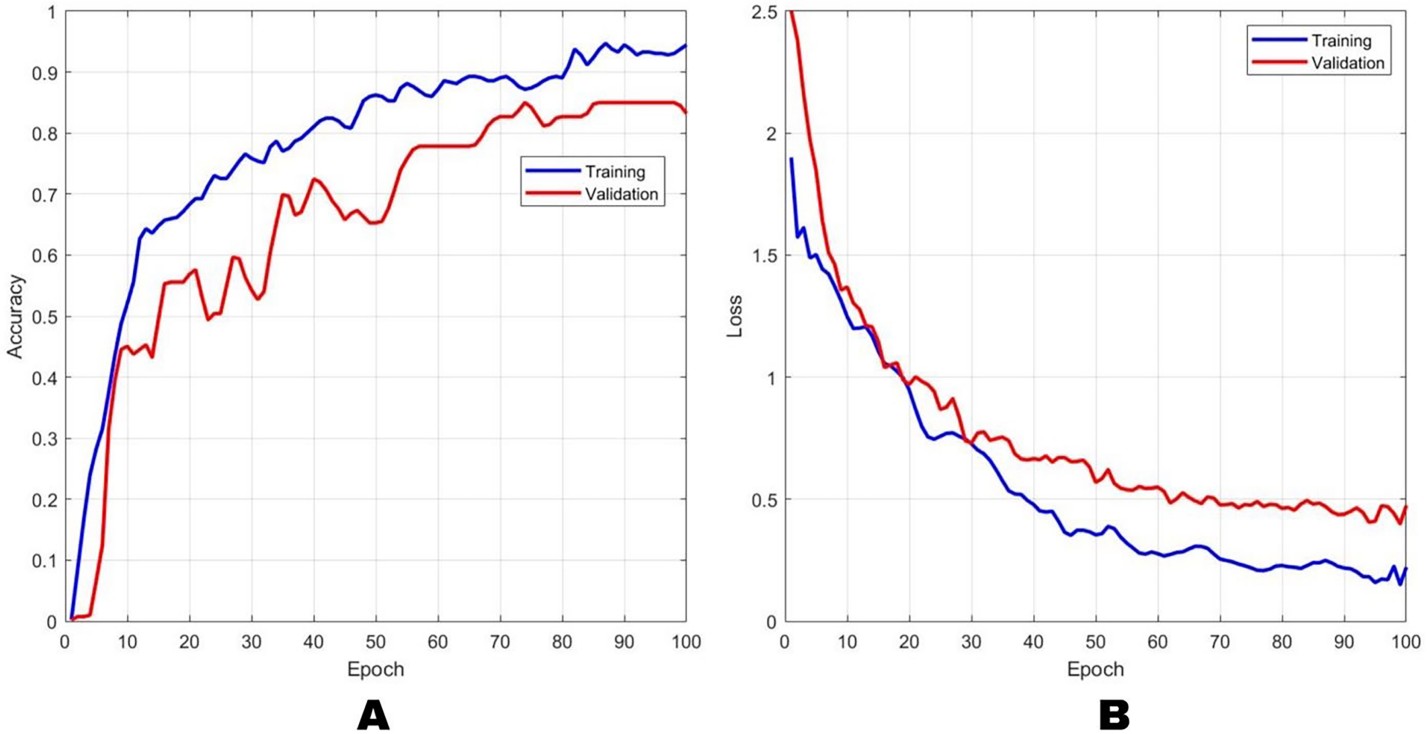

**Figure 5 The results of training and validation on the Block-based CS+AlexNet method for (A) accuracy and (B) loss.** The blue line indicates the training results, and the orange line indicates the testing results.

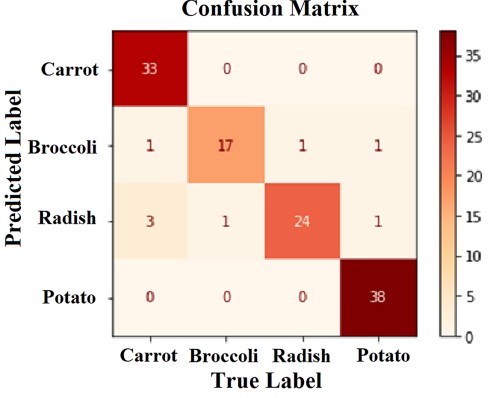

**Figure 6 Confusion matrix.**

Figure 7 shows a comparison of the computational time of each CR. Looking at the figure, it can be determined that the larger the CR, the greater the computation time, and the computation time without CS was the longest compared to that of the proposed combination technique.

The performance analysis of the proposed method compared with a previous study is shown in Table 4. Both proposed methods have higher accuracy performance than other

**Table 2 Precision, recall and F1-score values of the model to predict vegetables: carrot, brocolli, radish, potato.**

| Class | Precision | Recall | F1-score |
|---|---|---|---|
| Carrot | 0.96 | 0.88 | 0.92 |
| Broccoli | 1.00 | 0.93 | 0.96 |
| Radish | 1.00 | 0.94 | 0.97 |
| Potato | 0.82 | 0.97 | 0.89 |

**Table 3 Performance parameters of the proposed method.**

| Method | CR | Precision (%) | Recall (%) | F1-score (%) | Testing accuracy (%) | Testing loss (%) |
|---|---|---|---|---|---|---|
| Block-based CS in AlexNet | 32 | 81.25 | 83.50 | 82.35 | 81.66 | 52.02 |
| | 16 | 88.00 | 88.00 | 88.00 | 88.33 | 28.01 |
| | 8 | 92.25 | 93.00 | 92.62 | 92.50 | 17.35 |
| | 4 | 94.75 | 95.00 | 94.87 | 95.00 | 16.73 |
| | 2 | 96.25 | 96.25 | 96.25 | 96.66 | 14.90 |
| AlexNet | – | 97.45 | 98.00 | 97.55 | 98.00 | 15.00 |

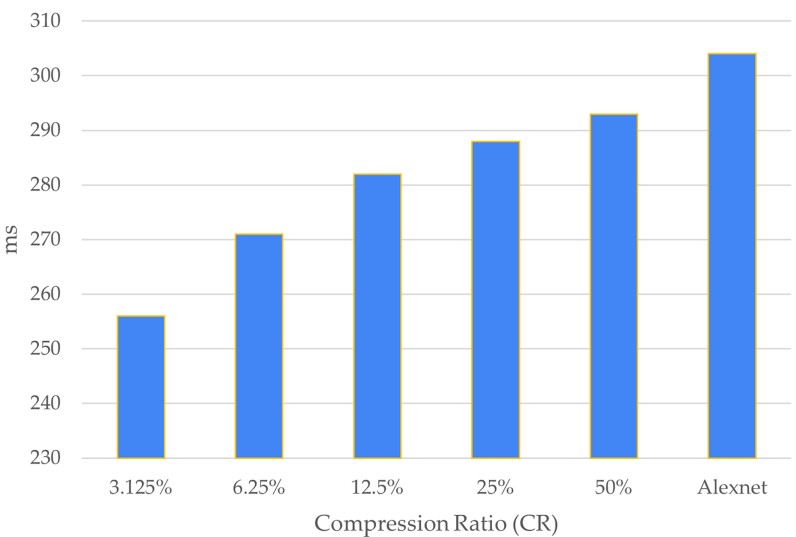

**Figure 7 Computational time.** The x-axis shows the compression ratio (CR) and the y-axis shows the times.

**Table 4 Performance comparison with previous study.**

| Data train | Data test | Accuracy | Method |
|---|---|---|---|
| 19,800 | 2,400 | 92.10% | AlexNet (*Zhu et al., 2018*) |
| 2,520 | 1,680 | 91.17% | AlexNet without augmentation (*Turaev, Almisreb & Saleh, 2020*) |
| 2,520 | 1,680 | 86.00% | AlexNet with augmentation (*Turaev, Almisreb & Saleh, 2020*) |
| 480 | 120 | 98.00% | AlexNet (proposed) |
| 480 | 120 | 96.66% | Block-based CS in AlexNet (proposed) |

previous studies. *Zhu et al. (2018)* used more than 23,200 images and *Turaev, Almisreb & Saleh (2020)* used 4,200 images, while only 600 images were used in this study. However, even with less data, our testing accuracy was higher. For our next research, we plan to optimize the CNN architecture after CS processing by several procedures, such as early stopping and over-fitting reduction as researcher suggestion from *Naushad, Kaur & Ghaderpour (2021)*.

## CONCLUSION

In this study, the CNN architecture AlexNet was used for vegetable image classification. Comparison tests were carried out limited on the AlexNet system only and AlexNet combined with block-based CS. In the CS system, DCT sparsity technique, Gaussian normal measurement matrix, and OMP reconstruction algorithm were used. Both proposed methods were tested on 600 datasets for four types of vegetables. The maximum testing accuracy results reached 98% for the AlexNet method, while the block-based CS in AlexNet method achieved a maximum accuracy of 96.66% with 2× compression rate. This proved that the two proposed systems provided better accuracy than previous studies. Future studies need to explore the addition of plant species and implementation when using sensors.

## ACKNOWLEDGEMENTS

We thank A. V. Senthil Kumar from Hindusthan College of Arts and Science, India for the comments on the manuscript.

### Funding

This research was funded by Telkom University through the 2022 International research scheme (No: KWR4.072/PNLT3/PPM-LIT/2022) in collaboration with University of Putra Malaysia. The funders had no role in study design, data collection and analysis, decision to publish, or preparation of the manuscript.

### Grant Disclosures

The following grant information was disclosed by the authors:
Telkom University through the 2022 International Research Scheme: KWR4.072/PNLT3/PPM-LIT/2022.
University of Putra Malaysia.

### Competing Interests

The authors declare that they have no competing interests.

### Author Contributions

- Indrarini Dyah Irawati conceived and designed the experiments, analyzed the data, prepared figures and/or tables, authored or reviewed drafts of the article, and approved the final draft.

- Gelar Budiman conceived and designed the experiments, performed the computation work, authored or reviewed drafts of the article, and approved the final draft.
- Sofia Saidah performed the computation work, authored or reviewed drafts of the article, and approved the final draft.
- Suci Rahmadiani performed the experiments, prepared figures and/or tables, and approved the final draft.
- Rohaya Latip analyzed the data, authored or reviewed drafts of the article, and approved the final draft.

## Data Availability

The code is available at GitHub and Zenodo:

- https://github.com/gelar1978/CS_vegetables_Alexnet/tree/main/CS_vegetables_Alexnet.

- Irawati, Indrarini Dyah, Budiman, Gelar, Saidah, Sofia, Rahmadiani, Suci, & Latip, Rohaya. (2023). Block-based compressive sensing in deep learning using AlexNet for vegetable classification (Final). Zenodo. https://doi.org/10.5281/zenodo.8358537.

The data are available at: M. I. Ahmed, S. Mahmud Mamun and A. U. Zaman Asif, "DCNN-Based Vegetable Image Classification Using Transfer Learning: A Comparative Study," 2021 5th International Conference on Computer, Communication and Signal Processing (ICCCSP), Chennai, India, 2021, pp. 235-243, DOI 10.1109/ICCCSP52374.2021.9465499.

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
