# Peer review of "Block-based compressive sensing in deep learning using AlexNet for vegetable classification"

_PeerJ Computer Science, doi:10.7717/peerj-cs.1551_

## Round 0.1 · original submission · Major Revisions

Dear Authors,

Your manuscript has been revised. According to the reviewers' comments, it needs major revisions before being considered for publication in this journal. You must improve both the description of your contribution and the description of the proposed approach. Especially the experiments need to be more detailed so the reader can replicate them.

Reviewer 1 ·

Basic reporting

Reviewer’s Report on the manuscript entitled:

Block-based Compressive Sensing in Deep learning using Alexnet for vegetable classification


The authors proposed a vegetable classification model that uses the CNN Alexnet model and compressive sensing to reduce computing time and save storage space. They compared the performance of their method with some other methods and showed their methods have better classification performance. The manuscript is interesting, but the presentation can be improved. Please see below my comments.

The literature review can be further improved by including some most recent articles:

Line 51. Please define ANN first and provide some references that shows its application for classification. For example, the following article that uses ANN for crop disease could be added here as an application: https://doi.org/10.3390/rs14051239

Line 55. Please define VGG. All the abbreviations must be defined the first time they appear. Also, please here add the following article that compares VGG with ResNet architectures for image classification: https://doi.org/10.3390/s21238083
The authors of the article above use techniques, such as early stopping to improve the computational efficiency and reduce over-fitting issues. Did you use such techniques in your proposed method? Either way, please discuss this at the end of the manuscript.

Line 82. Please define DCNN.

The following article also shows the applications of different deep learning architectures for vegetation image classification that can be added in the Introduction: https://doi.org/10.3390/signals3030031

Line 99. Please use bullet points to highlight the main contributions of this research.


Thank you for your contribution.

Regards,

Experimental design

Lines 108, 114, etc. please don’t use the letter x for multiplication. Please use the multiplication symbol instead. In latex it is “\times” that will create x.

Lines 138-173. Please add a flowchart showing these steps to help reader better understand your approach.

Table 1. It is “Negative” not “Negatif” and “Positive” not “Positif”.

Validity of the findings

'no comment'

Additional comments

Table 6 is in fact Table 5. Please check and correct the numbering.

Figures 3 and 4. Please insert the x-axis and y-axis labels.

Figure 5 has a very poor resolution. Please note that the resolution of all images should be at least 300 dpi.

Line 233. Please also mention the limitations of your method.

Reviewer 2 ·

Basic reporting

The paper presents an approach for vegetable classification using deep learning. The topics of the paper are interesting. However, the quality of the work is overall low and not suitable for publication in the present form. More in detail:
1) The main contributions of the paper are not clear. The novelties of the paper are not clearly highlighted, especially with respect to similar works on the topic.
2) The quality of the results and the analysis of the results is not sufficient for a journal publication in the present form.
3) The quality of the figures should be improved.
4) The literature review does not provide a sufficient overview on the topic.

Experimental design

The description of the approach should be improved. It would be impossible for the reader to replicate the experiments.

Validity of the findings

The impact and the novelties of the work are not clearly addressed throughout the manuscript.

---

## Round 0.2 · Minor Revisions

Dear Authors,

Your paper has been reviewed and some minor revisions are needed before it can be accepted for publication. In order to improve your manuscript, please consider the following suggestions:

1. Enhance the highlighting of your main contribution to the paper.
2. Improve the quality of the figures used in your paper.
3. Correct any typos or errors that may have been overlooked.

We appreciate your efforts and look forward to reviewing the revised version of your manuscript.

Reviewer 1 ·

Basic reporting

I would like to thank the authors for addressing my comments.

Please remove Table 1 because it does not add any information and it is well-known. And in line 190 please remove the sentence where it says: "Table 1 shows the term confusion matrix."

Line 188. Please remove "in the flowchart". Please also carefully check the Figure order and their description.

Please carefully check the format/style of the references, e.g., authors last names followed by abbreviated first names, volume numbers, etc.

Please carefully proofread the manuscript.

Experimental design

no comment

Validity of the findings

no comment

Additional comments

no comment

Reviewer 2 ·

Basic reporting

The quality of the paper is still low. I suggest to better highlight the main contributions of the work as well as the main results.
The quality of the figures is still very low. Some figures have a low resolution, not sufficient for a journal publication.

Experimental design

No comment

Validity of the findings

No comment

---

## Round 0.3 · accepted · Accept

Dear Authors,
Your manuscript has been accepted for publication in PEERJ Computer Science. The comments of the reviewers who evaluated your manuscript are included in this letter. I ask that you make minor changes to your manuscript based on those comments, before uploading final files.

Reviewer 1 ·

Basic reporting

I would like to thank the authors for addressing my comments satisfactorily. I have some minor suggestions:

Line 29. "...a comparison ratio of 2x." please replace 'x' with 'times' or 'twice' or something like that.

Line 54. Please replace "In agriculture, In agriculture, ANN can be used..." with "In agriculture, ANN was utilized...".

Please mention the limitations of your method in the Conclusion section.

Please carefully proofread the manuscript.

Thank you

Experimental design

no comment

Validity of the findings

no comment

Additional comments

The figure quality and their font size can be further improved. Please ensure a minimum resolution of 300 dpi for your figures.

Reviewer 2 ·

Basic reporting

The paper was not sufficiently improved considering my previous comments and suggestions. The quality of the paper is too low for a journal publication.

Experimental design

The paper was not sufficiently improved considering my previous comments and suggestions. The quality of the paper is too low for a journal publication.

Validity of the findings

The paper was not sufficiently improved considering my previous comments and suggestions. The quality of the paper is too low for a journal publication.

Additional comments

The paper was not sufficiently improved considering my previous comments and suggestions. The quality of the paper is too low for a journal publication.